# Short-Term Prediction of Multi-Energy Loads Based on Copula Correlation Analysis and Model Fusions

**DOI:** 10.3390/e25091343

**Published:** 2023-09-16

**Authors:** Min Xie, Shengzhen Lin, Kaiyuan Dong, Shiping Zhang

**Affiliations:** School of Electric Power, South China University of Technology, Guangzhou 510641, China; linsz_scutep@163.com (S.L.); 202220114455@mail.scut.edu.cn (K.D.); 202121016682@mail.scut.edu.cn (S.Z.)

**Keywords:** feature identification and extraction, Copula analysis, multi-energy loads, model fusion

## Abstract

To improve the accuracy of short-term multi-energy load prediction models for integrated energy systems, the historical development law of the multi-energy loads must be considered. Moreover, understanding the complex coupling correlation of the different loads in the multi-energy systems, and accounting for other load-influencing factors such as weather, may further improve the forecasting performance of such models. In this study, a two-stage fuzzy optimization method is proposed for the feature selection and identification of the multi-energy loads. To enrich the information content of the prediction input feature, we introduced a copula correlation feature analysis in the proposed framework, which extracts the complex dynamic coupling correlation of multi-energy loads and applies Akaike information criterion (AIC) to evaluate the adaptability of the different copula models presented. Furthermore, we combined a NARX neural network with Bayesian optimization and an extreme learning machine model optimized using a genetic algorithm (GA) to effectively improve the feature fusion performances of the proposed multi-energy load prediction model. The effectiveness of the proposed short-term prediction model was confirmed by the experimental results obtained using the multi-energy load time-series data of an actual integrated energy system.

## 1. Introduction

The safety, stability, and economic operations of traditional power systems depend on the short-term forecasting of power load, which has been extensively studied [1,2]. Further, other forms of load (cooling, heat, and gas energy) have some established foundations for research, mostly based on the pertinent aspects of this type of load, to conduct a single load prediction work. However, comprehensive energy system load forecasting for integrated energy systems with diverse energy coupling characteristics is still in its nascent exploration stage. Multi-energy load forecasting is an advanced field that builds upon traditional load forecasting. It seeks to reveal the complex nonlinear patterns resulting from the interconnected interplay between historical load trends and various influencing factors. Additionally, this type of forecasting requires an understanding of energy-coupling transformations and accounting for stochastic elements, such as integrating renewable energy sources and adaptable user demand responses. Reference [3] employs a genetic algorithm to refine a wavelet neural network, which is subsequently applied to optimize the prediction of heating loads in centralized heating systems. Similarly, Reference [4] takes into account building attributes such as construction year, dimensions, heat recovery ventilation, and geographical location, along with meteorological and seasonal influences, to construct a model that can predict building heating load demands. Reference [5] introduces a high-precision forecasting model for daily natural gas consumption by fusing pattern decomposition fusion techniques with integrated learning methods. Reference [6] refines the fruit fly algorithm using simulated annealing and cross-factor optimization. This enhanced algorithm is then integrated with support vector machine models, resulting in an elevated accuracy in forecasting urban natural gas loads. Diversifying the scope to cooling load prediction, Reference [7] tackles the forecast of building cooling loads using a non-linear autoregressive neural network with optimized parameters and external inputs. Further innovations are explored in Reference [8], where an integrated learning framework is proposed. This framework combines empirical mode decomposition and deep belief networks to predict cooling loads.

While the aforementioned single-energy prediction models largely fulfill the forecasting needs for energy demands within independent energy systems, the trend is undoubtedly moving towards a unified forecasting of multi-energy loads in integrated energy systems. This trend accounts for the evolving processes of cascaded utilization, optimized allocation, and interdependence within energy synergistic systems [9]. Reference [10] employs data mining techniques to perform a joint prediction of cooling and heating demands within buildings. This approach is utilized to evaluate energy-saving potential through multiple prediction models. In [11], a framework is established using least squares support vector machines combined with multi-task learning. This framework comprehensively predicts electric, cooling, heating, and gas loads in a campus integrated energy system. Furthermore, in [12], an unsupervised deep belief network (DBN) is fused with a supervised multi-task regression layer. This combined model is tested and validated using an industrial park integrated energy system, showcasing the effectiveness of deep multi-task learning approaches.

With the proliferation of artificial intelligence concepts and technological advancements, machine learning methods have found extensive application in the field of short-term load forecasting, particularly techniques like neural networks and support vector regression [13,14,15,16]. These methods are rapidly advancing due to their exceptional data mining abilities and their ability to tackle complex nonlinear problems. However, when applying these methods to short-term load forecasting, certain challenges may arise: (1) The enhancement of load information collection devices results in denser temporal data. Concurrently, the number of data sources affecting short-term load forecasting is increasing, including factors like weather, day types, economics, and societal influences. Therefore, when training models for short-term load forecasting, large input vectors could lead to diminished computational efficiency. (2) Models based on historical load data may exhibit significant errors when predicting load curve features, especially at peaks and valleys. This not only hampers the overall enhancement of predictive accuracy but may also result in the loss of crucial information, adversely affecting both the stability of the power grid and the long-term development of the electricity market.

The Douglas–Peucker (DP) algorithm, which is a classical method of curve feature extraction and compression, has advantages such as a high computational efficiency and strong visibility. Thus, it is appropriate for the curve feature extraction and dimensionality reduction in the multi-energy loads. However, a challenge intrinsic to the DP algorithm involves the judicious selection of an appropriate threshold. The threshold dictates the maximal distance, or error, between the approximate curve and the original curve. Within the context of multi-energy load forecasting, the application of the DP algorithm to primary load curve feature extraction and data compression exhibits considerable promise. Nonetheless, for optimal adaptation to distinct energy load curve nuances, prudent adjustments to and refinements of the threshold are needed, catering to the specific requirements of the analysis. Additionally, compared to the short-term prediction of a single energy load, the computational complexity of the short-term multi-energy load prediction is considerably higher. Hence, simple machine learning models are not suitable for the effective performance of short-term multi-energy load prediction tasks. The concept of classical fusion in deep learning can further improve the performance of the multi-energy load prediction systems.

The intricate coupling relationships among multi-energy loads significantly impact the accuracy and performance of short-term multi-energy load forecasting. By adapting and enhancing load feature recognition and combined forecasting models based on fixed patterns, a fuzzy-optimized load feature recognition and combined forecasting model better captures the precise characteristics of multi-energy loads, thus enhancing forecasting accuracy. In the context of the evolving comprehensive energy system development, the proposed fuzzy-optimized load feature recognition and combined forecasting model effectively addresses diverse and specific requirements for multi-energy load forecasting.

Motivated by the aforementioned facts, this paper proposes a new short-term prediction method for multi-energy loads based on the copula correlation feature analysis and model fusion. The main contributions of the paper are summarized as follows:(1)A two-stage approach to load feature identification and extraction is proposed. To address the challenges associated with the cumbersome and intricate threshold selection in the conventional DP algorithm, which is difficult to quantify and necessitates adaptive adjustments for different original datasets, the DP algorithm is improved by a fuzzy optimization threshold. After the initial feature extraction, the concept of statistical frequency distribution is applied to perform a secondary extraction of the collective characteristics of this load curve cluster to enhance the process of load feature identification and extraction.(2)Through the utilization of dynamically optimized Copula correlation measures, the input feature set of the multi-energy short-term forecasting model can be expanded. This integration ensures the thorough inclusion of interrelated characteristics among multi-energy loads into the predictive model, thereby effectively supplementing the model’s input information.(3)A multi-energy load forecasting model based on a model fusion framework is proposed. A Bayesian regularization (BR)-NARX (BR-NARX) neural network is used for the first prediction step, which uses BR to further optimize the performance of the traditional NARX model. Subsequently, a secondary forecasting model builds on the output of the primary model, utilizing a GA-optimized extreme learning machine (ELM) for separate multi-energy, short-term predictions of electricity, heat, and cooling loads. This approach ensures the comprehensive exploration of multi-energy load characteristics and elevates the accuracy of multi-energy, short-term load forecasting.

The remainder of this paper is organized as follows. Section 2 discribes the materials and methods used in the text, including a two-stage fuzzy-optimized load feature identification and extraction method, a multi-energy load correlated feature analysis based on the Copula method, and the construction of a multi-energy short-term forecasting framework through model fusion. Numerical case studies are discussed in Section 3. Section 4 concludes.

## 2. Materials and Methods

### 2.1. Two-Stage Optimization Method for Features and Extraction for Multi-Energy Loads

In this study, to select the key features of the multi-energy loads, we combined a fuzzy C-means (FCM) [17] with a two-stage fuzzy-improved Douglas–Peucker (TFIDP) algorithm. This method includes a three-step process that can be applied to the feature recognition and extraction of a load. The first step comprises performing FCM clustering on the multi-energy loads. In the second step, based on similar load curves, the DP algorithm improved by a fuzzy optimization threshold performs the initial feature extraction of the load. Finally, by exploiting the concept of statistical frequency distribution, a second feature extraction process is implemented based on the primary feature extraction.

#### 2.1.1. Initial Feature Extraction Based on a Fuzzy Optimization-Enhanced DP Algorithm

The classic DP algorithm extracts the feature points of a curve by setting the threshold value in advance. Further, the algorithm iteratively compares the vertical distance between the points of the updated target curve, first and last points, and set threshold size [18,19,20,21]. However, in practical applications, the threshold value needs to be specified using complicated factors that are difficult to quantify. Alternatively, the threshold value should be adjusted adaptively for different original datasets. Hence, it was essential to include an adaptive threshold to improve the TFIDP model.

The threshold value ε in the classical DP algorithm is usually set in the interval [0, 1] based on experience. For a series of curves with similar shape features, a reasonable threshold value must be set based on practical requirements. Hence, the DP algorithm threshold value can be set using fuzzy mathematics, which describes and models fuzzy concepts accurately to properly solve realistic problems. To improve the classical DP algorithm using the concept of fuzzy mathematics, we introduced a fuzzy optimized threshold value ε in the DP algorithm, which is the fundamental control standard for the final feature set extraction of curves. Further, by using the fuzzy mathematical concepts to optimize ε, the curve feature recognition and extraction process performed by the DP algorithm may improve in terms of generalizability.

The threshold value domain is defined as E∈[0,1] for the DP feature extraction algorithm. To simplify the operation, the threshold can assume a value in the range [0, 1] at discrete intervals of 0.1. $Sat\left(\varepsilon \right)$ represents the membership degree of the threshold value of the DP algorithm for a cluster of similar curves and is expressed as follows:(1)$$\left\{\begin{array}{l} Sat\left(\varepsilon \right)=a*D\left(\varepsilon \right)+b*Z\left(\varepsilon \right)\\ a+b=1 \end{array}\right.$$
where $D\left(\varepsilon \right)$ is the average matching degree between the curve features identified and extracted from the specified similar curve cluster and original curve, thereby reflecting the similarity between them. $Z\left(\varepsilon \right)$ is the percentage value of the number of curve feature points divided by the number of original curve points, which is the average percentage ratio of the original curve extracted and compressed using curve features. *a* and *b* are the corresponding proportion coefficients. The threshold value membership degree $Sat\left(\varepsilon \right)$ in Equation (1) comprises the sum of two parts and can be regarded as the overall curve feature extraction satisfaction for the specified similar curve cluster for a certain threshold value.

##### Average Matching Degree $D\left(\varepsilon \right)$

Because the time dimension of the curve features is reduced compared to that of the original curve, it is no longer a one-to-one mapping relationship. Thus, we introduced the dynamic time warping (DTW) algorithm to calculate the matching degree between the curve features and original curve [22]. DTW is often used in speech recognition tasks to measure the similarity between two time series with different lengths by calculating the DTW distance.

We define the original and characteristic sequences of the curve as X and Y, respectively, with corresponding sequence lengths $L_{X}$ and $L_{Y}$, and the warped length $w=[w_{1},w_{2},\cdots ,w_{K}]$, where $w_{i}=\left(p_{i},q_{i}\right)\in \left[1:L_{X}\right]\times \left[1:L_{Y}\right],1\leq i\leq K$. To satisfy the continuity and monotonicity at the boundary of the structured path, the main constraints are expressed as follows:(2)$$\left\{\begin{array}{l} w_{1}=\left(1,1\right),w_{K}=(L_{X},L_{Y})\\ p_{i+1}-p_{i}\leq 1,q_{i+1}-q_{i}\leq 1\\ p_{1}\leq p_{i}\leq p_{L_{X}},q_{1}\leq q_{i}\leq q_{L_{Y}} \end{array}\right.$$

The cumulative distance $F_{w}(X,Y)$ of the warped length between the original sequence *X* and feature sequence *Y* can be calculated using Equation (3).
(3)$$\left\{\begin{array}{l} d(x_{p},y_{q})={\left(x_{p}-y_{q}\right)}^{2}\\ F_{w}(X,Y)=\sum_{i=1}^{K}d(x_{p_{i}},y_{q_{i}}) \end{array}\right.$$

The minimum value of the cumulative distance is reached for the optimal warped length $w^{*}$ and corresponds, in this case, to the DTW distance $D_{dtw}$, which can be expressed as follows:(4)$$\left\{\begin{array}{l} w^{*}=\arg\min F_{w}(X,Y)\\ D_{dtw}(X,Y)=F_{w^{*}}(X,Y) \end{array}\right.$$

Based on the concept of dynamic programming, the cumulative distance of the optimal warped length can be calculated recursively using Equation (5). Here, the value of $D_{dtw}(x_{L_{X}},y_{L_{Y}})$ calculated iteratively is equal to the DTW distance between *X* and *Y.*
(5)$$\begin{array}{ll} D_{dtw}(x_{i},y_{j})= & d(x_{i},y_{j})+\min(D_{dtw}(x_{i-1},y_{j}),\\ & \;D_{dtw}(x_{i},y_{j-1}),D_{dtw}(x_{i-1},y_{j-1})) \end{array}$$

In the limit case, the curve features only include the first and last points of the original curve. Here, we denote the line connecting the first and last points of the original curve as $Y^{0}$. Further, the $D\left(\varepsilon \right)$ between the curve features identified and extracted from the similar curve cluster and original curve is calculated as follows:(6)$$D\left(\varepsilon \right)=\frac{\sum_{i=1}^{n}\left(1-\frac{D_{dtw}\left(X_{i},Y_{\varepsilon ,i}\right)}{D_{dtw}\left(X_{i},Y_{\varepsilon ,i}^{0}\right)}\right)}{n}\times 100\%$$
where *n* is the number of curves contained in the current similar curve cluster.

##### Average Compression Ratio $Z\left(\varepsilon \right)$

The average compression ratio $Z\left(\varepsilon \right)$ between the curve features identified and extracted from the similar curve cluster and original curve is calculated as follows:(7)$$Z\left(\varepsilon \right)=\left[1-\sum_{i=1}^{n}\left(\frac{Num\left(Y_{\varepsilon ,i}\right)}{Num\left(X_{i}\right)}\right)/n\right]\times 100\%$$
where the function $Num\left(\cdot \right)$ represents the amount of data in the obtained sequence.

##### Proportion Coefficients a and b

The selection of the proportion coefficients a and b is related to the importance of the terms $D\left(\varepsilon \right)$ and $Z\left(\varepsilon \right)$ in Equation (1). In this study, we set a = 0.7 and b = 0.3.

#### 2.1.2. Secondary Feature Extraction Based on Statistical Frequency Distribution

After the DP algorithm based on the fuzzy-optimized threshold completes the initial extraction of the characteristics of all the load curves in a certain load curve cluster, it applies the concept of statistical frequency distribution to extract the overall characteristics of this type of load curve cluster twice. All the non-repeated load characteristics, which are generated in the process of the load feature identification and extraction from a load curve cluster, are denoted as $I=[I_{1},I_{2},\cdots ,I_{i},\cdots ,I_{m}]$, with corresponding frequencies $G=[g_{1},g_{2},\cdots ,g_{i},\cdots ,g_{m}]$. The statistical frequency $f_{i}$ of each load characteristic for each load curve cluster can be calculated using Equation (8). Equation (9) is used to assess whether this load feature is suitable as one of the overall characteristics for the selected type of load curve cluster. If Equation (9) is satisfied, the corresponding $I_{i}$ is added to the feature number set $I^{\prime }$ of the updated load curve cluster, that is, $I_{i}\in I^{\prime }$.
(8)$$f_{i}=\frac{G_{i}}{I_{i}}$$
(9)$$f_{i}\geq 0.8\times \left(\sum_{i=1}^{m}f_{i}/m\right)$$

Finally, the set $I^{\prime }$ obtained using the TFIDP, exploiting the properties of the statistical frequency distribution, is used as the representative feature of the selected load curve cluster.

### 2.2. Analysis of Multi-Energy Load Correlation Characteristics Based on the Copula Method

To improve the final prediction accuracy of the short-term multi-energy load forecasting, several aspects must be considered, including the characteristics of the internal load represented by the historical development laws of the multi-energy loads, coupling conversion relationship of the load, characteristics of the external load, and meteorological aspects. Compared to the number of studies on the historical development law of the multi-energy loads and the correlation between the multi-energy loads and meteorological factors, only a few studies address the characteristics of the complex and flexible coupling conversion of the multi-energy loads. Hence, more accurate characterization methods are needed to effectively optimize the overall performance of the proposed short-term multi-energy load forecasting model. In this study, copula theory is used to model and analyze the correlation characteristics of the non-linear coupling conversion of the time series of multi-energy loads.

#### 2.2.1. Definition of the Copula Function

The copula theory was developed to solve a joint distribution problem of random variables when their marginal distributions are known. The formulation of Sklar’s theorem introduced the concept of the copula function. Further, models based on the copula function have been widely used in finance and economics, new energy output characteristic analysis, and other fields owing to their ability to explain complex nonlinearities among variables [23,24].

Sklar’s theorem proves that the joint distribution function of multiple n-dimensional variables can be constructed by combining the marginal distribution function of these variables and the associated copula function, which represents the complex correlation among variables. Considering an n-dimensional variable $x_{1},x_{2},\cdots ,x_{n}$ with a marginal distribution function in each dimension $F_{1}(x_{1}),F_{2}(x_{2}),\cdots ,F_{n}(x_{n})$, the joint distribution and density functions of the n-dimensional variable can be expressed as follows:(10)$$F(x_{1},x_{2},\cdots ,x_{n})=C[F_{1}(x_{1}),F_{2}(x_{2}),\cdots ,F_{n}(x_{n})]$$
(11)$$f(x_{1},x_{2},\cdots ,x_{n})=c(f(x_{1}),f(x_{2}),\cdots ,f(x_{n}))\cdot \prod_{i=1}^{n}f_{i}(x_{i})$$
where F is the joint distribution function of the n-dimensional variable, C the copula distribution function representing the complex correlation between n-dimensional variables, $F_{i}(x_{i})$ the density function of the variable in the *i*th dimension, f the joint density function of the n-dimensional variable, and c the copula density function.

Based on Equations (10) and (11), the copula density function can be obtained by taking the derivative of the copula distribution function, as expressed in Equation (12).
(12)$$c(f(x_{1}),f(x_{2}),\cdots ,f(x_{n}))=\frac{\partial^{n}C[F_{1}(x_{1}),F_{2}(x_{2}),\cdots ,F_{n}(x_{n})]}{\partial F_{1}(x_{1})\partial F_{2}(x_{2})\cdots \partial F_{n}(x_{n})}$$

By calculating the copula distribution and density functions, the complex correlation between the multivariate random variables can be accurately described. Typical static copula distribution functions include N-, T-, Gumbel, Clayton, and Frank copula functions. Additionally, the dynamic N-, T-, Clayton, and SJC copula functions can be used to model dynamic copula distribution functions. Hence, the correlation characteristics of the random variables can be analyzed using different copula distribution functions, which can reflect different perspectives.

#### 2.2.2. Correlation Analysis Based on Copula Functions

Generally, it is difficult to obtain a clear marginal distribution function because of the complexity of the multi-energy load time-series data. Hence, we applied the maximum likelihood estimation based on nonparametric kernel density (MLK) to estimate the correlation coefficient of the copula function [25,26]. The MLK method is not limited by the exact expression of the marginal distribution function. Instead, it uses the nonparametric kernel density estimation function of the analyzed variables.

For the normalized variable sequences *L_p_* and *L_c_*, the corresponding nonparametric kernel density estimation was conducted using the following equations:(13)$$f_{P}(x_{1})=\frac{1}{T}\sum_{t=1}^{T}K_{w}(x_{1}-L_{P})$$
(14)$$f_{C}(x_{2})=\frac{1}{T}\sum_{t=1}^{T}K_{w}(x_{2}-L_{C})$$
where $f_{P}(x_{1})$ and $f_{C}(x_{2})$ represent the probability density functions of $L_{P}$ and $L_{C}$, respectively. $K_{w}$ is the kernel function and *T* the size of the sequence of variables.

Using the MLK method, the static correlation coefficients can be obtained by substituting the marginal distributions with the probability density functions $f_{P}(x_{1})$ and $f_{C}(x_{2})$ in the likelihood function expressed in Equation (15). Further, its extreme points can be calculated using Equation (16).
(15)$$S(\theta )=\sum lnc[f_{P}(x_{1}),f_{C}(x_{2})]$$
(16)$$\hat{\theta }=argmaxS(\theta )$$

However, to calculate the dynamic time-varying correlation coefficient series, a dynamic copula function must be considered. Further, the likelihood function must be obtained using the parameters of the dynamic distributions associated with the variables and corresponding evolution equations. Hence, we define the dynamic N-copula function using the dynamic distribution parameter $\rho_{N,t}$ and dynamic T-copula function with parameters $\rho_{T,t}$ and $k_{t}$ degrees of freedom. The correlation coefficient matrices of the dynamic N- and T-copula functions based on the DCC (1,1) decomposition can be expressed as follows:(17)$$R_{t}={\left(Q_{t}^{*}\right)}^{-1/2}\cdot Q_{t}\cdot {\left(Q_{t}^{*}\right)}^{-1/2}$$
where $Q_{t}^{*}=diagQ_{t}$ and its evolution equation is expressed as follows
(18)$$Q_{t}=R(1-\alpha -\beta )+\alpha (\varepsilon_{t-1}\varepsilon_{t-1}^{\prime })+\beta Q_{t-1}$$
where α and β represent the estimated evolution parameters, which satisfy the constraints 0<α<1, 0<β<1 and 0<α+β<1, respectively. $\varepsilon_{t}$ is the pseudo-inverse of the threshold distribution function.

The evolution equation of the dynamic Clayton copula function is defined as follows:(19)$$\theta_{C,t}=\Lambda (\omega +\beta \theta_{t-1}+\alpha \cdot \frac{1}{10}\sum_{j=1}^{10}\left(|L_{P,t-j}-L_{C,t-j}\right|)$$
where ω, α, and β represent the estimated evolution parameters. $\Lambda (x)=\frac{\left(1-e^{-x}\right)}{\left(1+e^{-x}\right)}$ is the restriction function.

The evolution equation of the dynamic SJC copula function is expressed as follows:(20)$$\tau_{t}^{U}=\tilde{\Lambda }(\omega_{U}+\beta_{U}\tau_{t-1}^{U}+\alpha_{U}\cdot \frac{1}{10}\sum_{j=1}^{10}\left|L_{P,t-j}-L_{C,t-j}\right|)$$
(21)$$\tau_{t}^{L}=\tilde{\Lambda }(\omega_{L}+\beta_{L}\tau_{t-1}^{L}+\alpha_{L}\cdot \frac{1}{10}\sum_{j=1}^{10}\left|L_{P,t-j}-L_{C,t-j}\right|)$$
where $\tau_{t}^{U}$ is the upper-tail dependence coefficient, $\omega_{U}$, $\alpha_{U}$, and $\beta_{U}$ the estimated parameters of the upper-tail evolution, $\tau_{t}^{L}$ the lower-tail dependence coefficient, and $\omega_{L}$, $\alpha_{L}$, and $\beta_{L}$ the estimated parameters of the lower-tail evolution. The restriction function $\tilde{\Lambda }(x)$ satisfies $\tilde{\Lambda }(x)=(1+e^{-x}{)}^{-1}$.

In this study, AIC was used to evaluate the adaptability of the different copula models presented above. The AIC index is calculated as follows:(22)AIC=2k−2lnS
where *k* represents the number of parameters of the copula function and *S* the associated maximum likelihood estimation.

#### 2.2.3. Analysis of Multi-Energy Load Characteristics Based on Copula Methods

The steps of the analysis of the multi-energy loads using the copula method are summarized in Table 1.

Copula functions can accurately describe the complex non-linear coupling relationship characterizing the multi-energy loads, which can considerably improve the overall performance of the proposed short-term multi-energy load prediction model. As shown in Figure 1, the copula function-based feature analysis process proposed in this study includes the following steps:Multi-energy load data are normalized in the [0, 1] interval to ensure data uniformity.The kernel density estimation function is calculated using the MLK method, which determines the marginal density function of the variable sequence.To obtain static correlation coefficients, the marginal density copula functions are used to calculate the extreme points of the likelihood function.To obtain dynamic correlation coefficients, the dynamic copula distributions are used to construct the likelihood function considering the corresponding evolution equation parameters (ω, α, and β).Once the maximum likelihood estimates and the corresponding evolution equation parameters are obtained, they are substituted into the evolution equation parameters to calculate the required time-varying cross-correlation coefficients.Simultaneously, the selected copula functions are optimized based on the maximum likelihood estimate, and later the optimal copula model is obtained by comparing the corresponding AIC indexes.

### 2.3. Short-Term Forecasting Framework for Multi-Energy Loads Based on Model Fusion

The copula function-based analysis can effectively capture the intricate non-linear coupling between multi-energy loads. Using the optimal copula correlation measure, the input feature set of the proposed short-term prediction model for multi-energy loads can be enhanced with the interrelated characteristics of the multi-energy loads.

To further improve the performance of multi-energy load forecasting, we developed an approach that exploits model fusion in this study. For the first prediction step, we introduced a Bayesian regularization NARX (BR-NARX) neural network to predict the characteristics of the electrical, heating, and cooling loads. Based on the output of this primary prediction model, the secondary prediction is obtained using a GA-optimized ELM model that returns the final short-term prediction of electricity, heat, and cooling loads. Owing to this two-step process, the characteristics of the multi-energy loads are fully explored, enhancing the accuracy level of the proposed short-term prediction model.

#### 2.3.1. BR-NARX Model

Owing to its reasonable structural performance, the NARX neural network model effectively captures the nonlinearity of the time series. Further, its parallel distribution training mode improves fault tolerance and stability, making this model more competitive than other typical machine learning approaches [27,28]. In this study, we introduced BR to further optimize the performance of the traditional NARX model.

Traditional neural network models often adopt the backpropagation algorithm to adjust network parameters during the training process. Here, the error performance function $E_{d}$ is usually defined as the sum of the mean squared errors, as expressed below:(23)$$E_{d}=\frac{1}{n}\sum_{i=1}^{n}{\left(t_{i}-p_{i}\right)}^{2}=\frac{1}{n}\sum_{i=1}^{n}{\left(e_{i}\right)}^{2}$$
where $t_{i}$ is the expected value of the ith actual target, $p_{i}$ the ith output value predicted by the neural network, $e_{i}$ the *i*th absolute error of prediction, and *n* the total number of input samples trained by the neural network. By introducing the regularization optimization method for weight coefficients and threshold parameters, the performance of the neural network models can be enhanced in terms of the limited overall parameter scale and improved generalization ability.

Based on the regularization optimization concept [29], the regularization optimization error performance function $E_{rgl}$ of the neural network is modified as follows:(24)$$E_{rgl}=\alpha E_{\omega }+\beta E_{d}$$
(25)$$E_{\omega }=\sum_{i=1}^{n}\omega_{i}^{2}$$
where $\omega_{i}$ is the weight coefficient of the neural network, $E_{\omega }$ the sum of the squares of weight coefficients, and α and β the regularization optimization parameters weighting the contribution of $E_{\omega }$ and $E_{d}$, respectively. The more $E_{\omega }$ is restricted, the stronger the generalization performance of the neural network will be. If α≫β, the purpose of neural network training is to limit the size of network parameters, which may result in large training errors. On the contrary, if α≫β, Equation (24) describes the typical mean squared error performance function, which may lead to overfitting. However, it is often difficult to practically determine the optimal size of the network parameters ω.

For specific neural networks aimed at real-world problems, determining the appropriate network parameter scale ω can often be challenging. BR optimization theory, rooted in Bayesian probability formulas, can leverage actual target expectations and Bayesian probability estimation to infer and analyze unknown regularization parameters α and β in a rational manner. The BR optimization framework assumes that the weight coefficients of the neural network are stochastic variables. Given a training dataset *D* and the neural network’s structural form *M*, the posterior probability of the neural network’s weight coefficients can be derived using Bayesian probability formulas as follows:(26)$$P(\omega |D,\alpha ,\beta ,M)=\frac{P(D|\omega ,\beta ,M)P(\omega |\alpha ,M)}{P(D|\alpha ,\beta ,M)}$$
where *ω* represents the weight coefficient vector of the neural network. *P*(*D*|*ω*,*β*,*M*) denotes the observed probability of the dataset *D* given the neural network’s weight coefficient vector *ω*. *P*(*ω*|*α*,*M*) is the prior probability of the network’s weight coefficients before considering the training data, given the structural form *M*. *P*(*D*|*α*,*β*,*M*) represents the validation probability of the dataset *D* in the neural network model with the given hyperparameters *α* and *β*, determined by Equation (27):(27)P(D|α,β,M)=∫P(D|ω,β,M)P(ω|α,M)dω

Assuming that the noise in the training dataset and the prior probability of network weight coefficients follow Gaussian distributions, we have:(28)$$\left\{\begin{array}{l} P(D|\omega ,\beta ,M)=\frac{1}{Z_{D}(\beta )}e^{-\beta E_{d}}\\ Z_{D}(\beta )={\left(\frac{\pi }{\beta }\right)}^{\frac{n}{2}} \end{array}\right.$$
(29)$$\left\{\begin{array}{l} P(\omega |\alpha ,M)=\frac{1}{Z_{W}(\alpha )}e^{-\alpha E_{d}}\\ Z_{W}(\alpha )={\left(\frac{\pi }{\alpha }\right)}^{\frac{n}{2}} \end{array}\right.$$
where *n* represents the number of samples in the training dataset *D*. Therefore, by using the equation above, we can derive:(30)$$\left\{\begin{array}{l} P(\omega |D,\alpha ,\beta ,M)=\frac{1}{Z_{E_{rgl}}(\alpha ,\beta )}e^{-E_{rgl}(\omega )}\\ Z_{E_{rgl}}(\alpha ,\beta )=Z_{D}(\beta )Z_{W}(\alpha )P(D|\alpha ,\beta ,M) \end{array}\right.$$

BR optimization theory aims to maximize the posterior probability of neural network weight coefficients to estimate the network parameter scale. Thus, minimizing *E*_rgl_(***ω***) is equivalent to maximizing *P*(***ω***|*D*,*α*,*β*,*M*), which means that *E*_rgl_(***ω***) reaches its minimal value at the point ***ω***_minP_. As for the regularization parameters, their posterior probability is represented as shown in Equation (31):(31)$$P(\alpha ,\beta |D,M)=\frac{P(D|\alpha ,\beta ,M)P(\alpha ,\beta |M)}{P(D|M)}$$

Assuming a uniform distribution for the prior probability *P*(*α*,*β*|*M*), maximizing the posterior probability *P*(*α*,*β*|*D*,*M*) of the regularization parameters is equivalent to maximizing *P*(*D*|*α*,*β*,*M*). From the previous equation, it can be deduced that:(32)$$P(D|\alpha ,\beta ,M)=\frac{Z_{E_{rgl}}(\alpha ,\beta )}{Z_{D}(\beta )Z_{W}(\alpha )}$$

Furthermore, due to the quadratic shape of the objective function of BR optimization near the minimization point *ω*_minP_, and the fact that its gradient is zero, it is possible to estimate it using a Taylor series expansion.
(33)$$Z_{E_{rgl}}(\alpha ,\beta )={\left(2\pi \right)}^{\frac{n}{2}}{\left[\det{\left(H_{\min P}\right)}^{-1}\right]}^{\frac{1}{2}}e^{-E_{rgl}(\omega_{\min P})}$$
(34)$$H=\alpha \cdot \nabla^{2}E_{\omega }+\beta \cdot \nabla^{2}E_{d}$$
where ***H*** represents the Hessian matrix of the objective function in BR optimization, and ***H***_minP_ is the Hessian matrix evaluated at the minimization point ***ω***_minP_.

Substituting the estimated value from Equation (33) into Equation (32), and taking the logarithm on both sides while setting the value to zero, the optimal Bayesian regularized optimization parameter at *ω*_minP_ can be obtained as follows [30]:(35)$$\left\{\begin{array}{l} \alpha^{*}=\frac{\gamma }{2E_{\omega }\left(\omega_{\mathrm{minP}}\right)}\\ \begin{array}{l} \beta^{*}=\frac{n-\gamma }{2E_{d}\left(\omega_{\mathrm{minP}}\right)}\\ \gamma =n-2\alpha^{*}\cdot \mathrm{tr}{\left(H_{\mathrm{minP}}\right)}^{-1} \end{array} \end{array}\right.$$
where $\alpha^{*}$ and $\beta^{*}$ represent the optimal regular optimization parameters, $\omega_{minP}$ the minimum point of the network weight coefficient, $H_{minP}$ the Hessian matrix of the BR objective function when the value of $H_{minP}$ is minimal, γ the number of effective parameters of the neural network, and tr(∙) the trace of the matrix. During the neural network training, the regularized optimization parameters α and β are dynamically adjusted based on the above BR approach, thereby implementing an adaptive learning method that improves the overall generalization ability of the neural network using a limited training dataset. After the objective function is defined, the Levenberg–Marquardt algorithm is used to minimize it.

#### 2.3.2. Combined Genetic Algorithm and Extreme Learning Machine Model

The ELM model introduces the concept of stochastic optimization in neural network applications. In this framework, the connection weights between the input and hidden layers and the bias values of the neurons in the hidden layers can be randomly generated. As opposed to the repeated training and adjustment that is typical of the gradient descent methods, the only parameter that needs to be set in the training of the ELM algorithm is the number of neurons of the hidden layer. The optimal solution under the corresponding conditions can be simply obtained by calculating the generalized inverse matrix, resulting in several application advantages. Hence, the introduction of ELM has considerable advantages in terms of the training performance. However, it is difficult to ensure that the optimal parameters are selected for the actual prediction under the influence of unknown features. Moreover, the generalization ability of the model needs to be further improved.

Figure 2 summarizes the above optimization process of the ELM model using GA. The obtained GA-ELM optimization model represents the output layer of the proposed short-term multi-energy load prediction model.

#### 2.3.3. Overall Modeling Framework

The complete framework of the proposed multi-energy load forecasting model, which was developed using copula-function-based feature analysis and model fusion, is shown in Figure 3.

(1) Feature Extraction: The electric, heating, and cooling load sequence data are subjected to feature extraction using the FCM-TFIDP method. Meteorological data, day type, and other load-influencing-factor data are extracted using PCA. The joint features are combined to form the input vector of the first-layer BR-NARX prediction model, which performs feature prediction for electric, thermal, and cooling load sequences. (2) Expansion of Features: Based on the output features of the first-layer BR-NARX prediction model for electric, thermal, and cooling loads, additional features derived from PCA-extracted meteorological and day type factors, as well as optimal Copula-dynamic-related features among multi-energy loads, are incorporated. These fused features become the input vector for the second-layer GA-ELM prediction model. (3) Model Fusion and Optimization: Through the fusion between the first-layer and second-layer prediction models and individual parameter optimization of each layer’s models, the optimal predictive model is derived. This process leads to the separate prediction of multi-energy load values for cooling, thermal, and electric loads.

This framework significantly enhances adaptability by combining load feature recognition with pattern-based predictive models. The fuzzy-optimized load feature recognition model improves the identification and prediction of precise features for multi-energy loads. The dynamic Copula-related feature analysis comprehensively captures complex nonlinear coupling relationships among multi-energy loads. By incorporating dynamically optimized Copula-related measures into the model, the input feature set of the multi-energy load short-term prediction model is expanded, effectively integrating correlated features between different types of energy loads. Furthermore, the framework considers meteorological, day type, and other influencing factors. It applies Bayesian regularization optimization and genetic algorithm optimization to improve the NARX and ELM prediction models in the two layers. By leveraging the model fusion framework, the predictive strengths of different model structures are efficiently harnessed, enhancing the overall model’s ability to learn and process fused features of multi-energy loads. This enhancement improves the practical performance of multi-energy load forecasting models in real-world applications.

## 3. Results and Discussion

The original data used for the experiment performed in this study were obtained from the operating load data of the integrated energy system in a similar actual park from August 2019 to October 2020. Particularly, we used the total daily curve data of the electrical, heating, and cooling loads in the energy supply area of the system. The sampling interval of the multi-energy load data was set to 15 min (the daily load curve comprised 96 load points acquired from 00:00 to 23:45). To simplify the analysis, the measurement units of the cooling and heating loads were converted into MW (the unit of the electrical load). The load-influencing factors mainly included the meteorological information and other daily varying information. The meteorological data generally include daily temperature, humidity, air pressure, wind direction and speed. The daily varying data include information on working and rest days, and holidays.

In the proposed framework, data preprocessing was performed to convert the units of the other loads to that of the electrical load. Here, data cleaning is performed on the original data. The data are then normalized to obtain values in the interval [0, 1]. Additional required data can be found in Table 2. The relevant content will be provided as an addition in the revised manuscript. For our experiment, we used a 3.00 GHz Intel Core I7 with 16 GB memory. The proposed model was implemented using MATLAB R2018b. This article references the comprehensive park data of Guizhou Power Grid Co., Ltd. (Guiyang, China)’s technological project: 061000KK52180003.

### 3.1. Copula-Related Characteristic Analysis Based on Multi-Energy Loads

Copula function-based characteristic analysis is considered from the perspective of multi-energy loads. Particularly, it is expected that the correlation between the multi-energy loads and external factors can be reasonably quantified through the optimal copula correlation coefficient under a certain measurement index. Consequently, the new characteristic connotation information can be introduced to provide a better reference, thus improving the accuracy of the proposed multi-energy load prediction model. To exploit the copula function correlation analysis, the optimal copula function between the electrical and cooling load series was selected among eight alternative copula functions based on the AIC criterion and maximum likelihood estimate values reported in Table 1.

The optimal copula function of the multi-energy loads is selected based on the AIC criterion and maximum likelihood estimate. Under optimal conditions, the AIC value of the optimal copula function must be as small as possible, whereas the maximum likelihood estimate value must be as large as possible. From Table 1, the optimal copula function of the electrical and cooling loads series based on the selected criteria is the SJC copula function. In fact, its AIC value of −3284.187 is the smallest, and its maximum likelihood estimate value of 1648.263 is the largest among the values of all the other copula functions. Hence, it is reasonable and efficient to choose the SJC copula function to analyze the correlation characteristics of the electrical and cooling loads series.

The above optimization analysis of the copula functions for the electrical–heating and heating–cooling load series shows that the optimal copula functions for the multi-energy load series based on the selected criteria are the SJC copula functions. The SJC copula function reflects the static correlation characteristics of the multi-energy loads and provides an improved representation of their dynamic correlation characteristics. By analyzing the main dynamic distribution parameters and evolution equations of the SJC copula function, it can be observed that the dynamic coefficient of the tail dependence reflects the dynamic relationship of the time series, which is suitable for the extended input of multi-energy load forecasting. Hence, it provides a reference for the prediction model that considers the coupling characteristics of the multi-energy loads.

### 3.2. Model Parameter

To improve the prediction accuracy of the final prediction model, the parameters of each sub-model were optimized. The key parameters of the proposed model are summarized in Table 3.

For the BR-BARX neural network model in the first layer, a trial optimization method was adopted to determine the number of neurons in the key hidden layer and delay order. For the GA-ELM model in the second layer, the number of neurons in the hidden layer was determined using a method that combines trial optimization and GA algorithms.

### 3.3. Evaluation of the Model Performance

The performance evaluation metrics of the proposed prediction model adopted in this study include the relative error rate $E_{i}$ at the *i*th point of the load prediction, root-mean-squared error $E_{RMSE}$ of total load prediction, and rate of the mean absolute error $E_{MAPE}$, accuracy of prediction Acc, overall mean absolute error of multi-energy loads $E_{SUMMAPE}$, and overall prediction accuracy of the multi-energy loads $Acc_{SUM}$. These metrics are defined in Equations (27)–(32), where ${\hat{x}}_{i}$ represents the predicted value of the electrical load at the ith point and $x_{i}$ the actual value of the electrical load at a similar point. $E_{MAPE,P}$, $E_{MAPE,H},$ and $E_{MAPE,C}$ represent the rate of the overall mean absolute error of the electrical, heating, and cooling loads, respectively. $Acc_{P}$, $Acc_{H}$, and $Acc_{C}$ represent the rate of the overall prediction accuracy of the electrical, heating, and cooling loads, respectively. $\omega_{P}$, $\omega_{H}$, and $\omega_{C}$ are the energy allocation proportion coefficients of the electrical, heating, and cooling loads, respectively, which satisfy the relationship $\omega_{P}+\omega_{H}+\omega_{C}=1$. In this study, the ratio coefficient of the electrical, heating, and cooling loads was set to 0.4:0.4:0.2 based on the actual energy configuration of the examined system.
(36)$$E_{i}=\frac{{\hat{x}}_{i}-x_{i}}{x_{i}}\times 100\%$$
(37)$$E_{\mathrm{RMSE}}=\sqrt{\frac{1}{n}\sum_{i=1}^{n}{\left({\hat{x}}_{i}-x_{i}\right)}^{2}}$$
(38)$$E_{\mathrm{MAPE}}=\frac{1}{n}\sum_{i=1}^{n}\left|E_{i}\right|\times 100\%$$
(39)$$Acc=\left(1-\sqrt{\frac{1}{n}\sum_{i=1}^{n}{\left(\frac{{\hat{x}}_{i}-x_{i}}{x_{i}}\right)}^{2}}\right)\times 100\%$$
(40)$$E_{\mathrm{SUMMAPE}}=\omega_{P}E_{\mathrm{MAPE},P}+\omega_{H}E_{\mathrm{MAPE},H}+\omega_{C}E_{\mathrm{MAPE},C}$$
(41)$$Acc_{\mathrm{SUM}}=\omega_{P}Acc_{P}+\omega_{H}Acc_{H}+\omega_{C}Acc_{C}$$

### 3.4. Results

The daily electrical, heating, and cooling loads in a typical week of September 2020 (13 September 2020.) were selected as the prediction objects. Using the copula function feature analysis and model fusion layer of the proposed short-term multi-energy load prediction model, the multi-energy load prediction was performed. To analyze the prediction performance of the proposed model, we applied three other models to the collected dataset to compare the multi-energy load predictions in the selected period. The first model (group 1) was obtained considering the modules TFIDP, PCA, and BR-NARX only. The second model (group 2) included a similar module to the first one with the addition of the copula function-based characteristic analysis of the multi-energy loads. The third model (group 3) was obtained by adding the model fusion method to the second model, with GA-ELM as the second layer of the prediction model. The fourth comparison model (group 4) was the complete short-term multi-energy load prediction model proposed in this study.

For the quantitative analysis, the metrics presented in Section 3 were used to evaluate the multi-energy load prediction performance of the comparison models. Further, to evaluate the differences in the prediction performance on weekdays and rest days, we performed a separate experiment using the four comparison models to objectively and comprehensively evaluate the utility of the model components.

From the evaluation of the multi-energy load prediction results of weekdays reported in Table 4, the overall effect on the predictions of the electrical and cooling loads is better than that of heating. In fact, the electrical and cooling loads often show a relatively stable evolution trend, while the heating load is driven by random energy demand and has the characteristics of time lag, thereby making the rule of change difficult to control.

Comparing the prediction results of groups 1–3 to those of the complete proposed model (group 4) confirmed that the copula-function-based, BR-NARX, and GA-ELM modules fully benefitted from the feature-extraction ability of the TFIDP–PCA method. Additionally, the copula correlation coefficient feature was introduced to expand the fusion feature, while the model fusion design was added simultaneously. The GA-ELM strong generalization ability was used for the analytic learning of fusion features. Thus, the prediction accuracy of the multi-energy load forecasting on weekdays is effectively improved.

The results reported in Table 5 show that, on rest days, the variation trend of the multi-energy loads is more random.

Hence, the accuracy of the multi-energy load prediction results on rest days is worse than that on weekdays. However, for the group 4 model, the values of the *E*_RMSE_ and *E*_MAPE_ of the cooling load prediction results were lower on rest days than on weekdays. To a certain extent, these results confirm that the proposed model has good generalization and strong anti-fluctuation abilities regardless of the type of day, which proves its practical effectiveness. Figure 4, Figure 5 and Figure 6 show the prediction results of the electrical, heating, and cooling loads of the integrated energy system considered in this study over one week for the four groups, further confirming that the complete prediction model of the multi-energy loads proposed in this study has the best performance. A high prediction accuracy was achieved for the electrical and cooling loads by all the groups owing to their relatively stable variation trend. The heating load, which is characterized by a more random variation trend compared to that of the other loads, was tracked more effectively by the full model, which provided improved prediction results compared to those obtained by groups 1–3.

By taking full advantage of the feature-extraction ability of the TFIDP–PCA method, the improved feature fusion based on the copula correlation coefficient, and characterization of the multi-energy load coupling conversion relationship, the proposed model ensures high-accuracy prediction on both working and rest days, as confirmed by our experimental results.

In order to further evaluate and compare the short-term prediction results of the multi-energy load proposed in this chapter and the control model, the prediction accuracy of various multi-energy load short-term prediction models is analyzed by using the absolute value of the relative error rate of electric, heat and cold loads in the form of box plot. As shown in Figure 7, Figure 8 and Figure 9, model 1 is TFIDP-PCA-BRNARX model, model 2 is Copula-TFIDP-PCA-BRNARX model, model 3 is Copula-TFIDP-PCA-BRNARX-ELM model and model 4 is the Copula-TFIDP-PCA-BRNARX-GAELM model proposed in this chapter. It can be seen from Figure 7 that, for the comparison of the prediction errors of the electric load, the mean value of the absolute value of the relative error rate of the model 4 proposed in this chapter is the smallest among all the prediction models, and the shape of the box is relatively flat, reflecting that the prediction error is at a small level. It is relatively more concentrated, so the overall prediction performance is the best. Meanwhile, by comparing the predictive errors of the thermal and cooling loads in Figure 8 and Figure 9 the application advantages of the model proposed in this chapter for short-term multi-energy load forecasting are similarly demonstrated.

From the analysis of the experimental results, we can conclude that the introduction of previous knowledge represents a key aspect of the improvement in the prediction accuracy of the multi-energy load forecasting. In fact, the multi-energy load characteristic correlation analysis considerably enhanced the multi-energy load prediction accuracy of the integrated energy system. Additionally, owing to the combined effect of the TFIDP–PCA method and BRNARX model, the proposed short-term prediction model effectively predicts the features of each load of the multi-energy integrated system independently.

## 4. Conclusions

In this paper, the key features of the multi-energy load curve are selected by the two-stage load feature recognition and extraction method, and the coupling nonlinear feature relationship between the multi-energy loads is quantified based on the Copula correlation analysis to introduce the Copula correlation analysis feature results and expand the feature input. At the same time, the model fusion framework is used to construct a short-term multi-energy load forecasting model with better prediction accuracy. Our experimental results confirmed that the copula correlation analysis could effectively quantify the coupling relationship of the multi-energy loads. Additionally, the time-varying copula correlation coefficient effectively enhanced the feature input of the multi-energy load prediction model by enriching the associated information content, thereby improving the prediction accuracy of the model. Lastly, by exploiting the model fusion, the advantages of the predictive models with different structures were effectively combined to improve the learning and processing ability in the multi-energy load feature fusion and generalization performance in practical applications of the proposed multi-energy load prediction model.

In follow-up studies, the following points should be considered:(1)At present, the existing short-term multi-energy load prediction research is mostly modeled from the perspective of a single energy form output, and in future comprehensive energy system multi-energy load forecasting research, the multi-objective prediction should be studied accordingly, so as to better link the coupling characteristics between multi-energy loads to improve the corresponding prediction effect.(2)The multi-energy load represented by the intelligent building building is transmitted, distributed and converted by the energy topology network architecture and energy coupling conversion device equipment in the park, so there are not only correlation characteristics at the time scale, but also correlation characteristics at the spatial scale, and the next step is to analyze the load characteristics from the perspective of spatiotemporal correlation to more accurately characterize the coupling conversion characteristics of multi-energy loads.(3)The existing short-term multi-energy load forecasting research has less analysis and less consideration from the perspective of multi-energy marketization. In the new environment of the development of a multi-energy market mechanism, how to consider the characteristics of multi-energy load brought by marketization will be the next meaningful research direction.

## Figures and Tables

**Figure 1 entropy-25-01343-f001:**
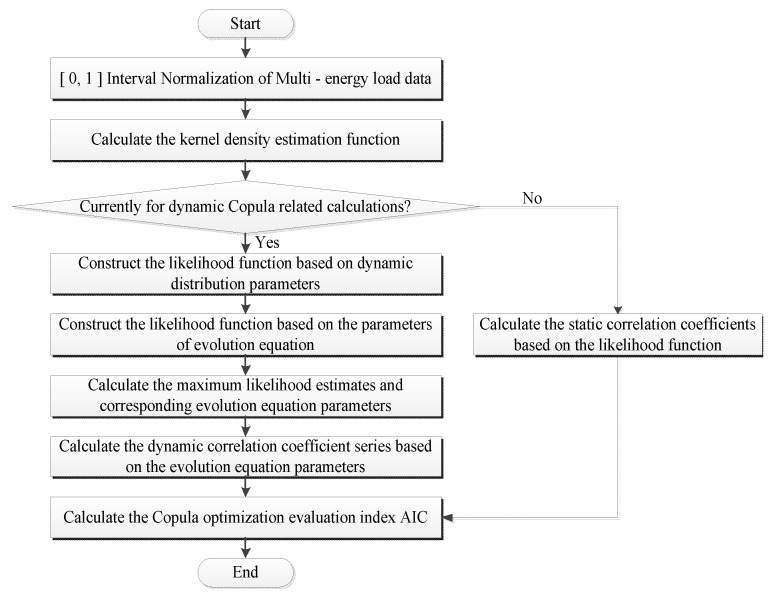
Schematic of the copula feature analysis process.

**Figure 2 entropy-25-01343-f002:**
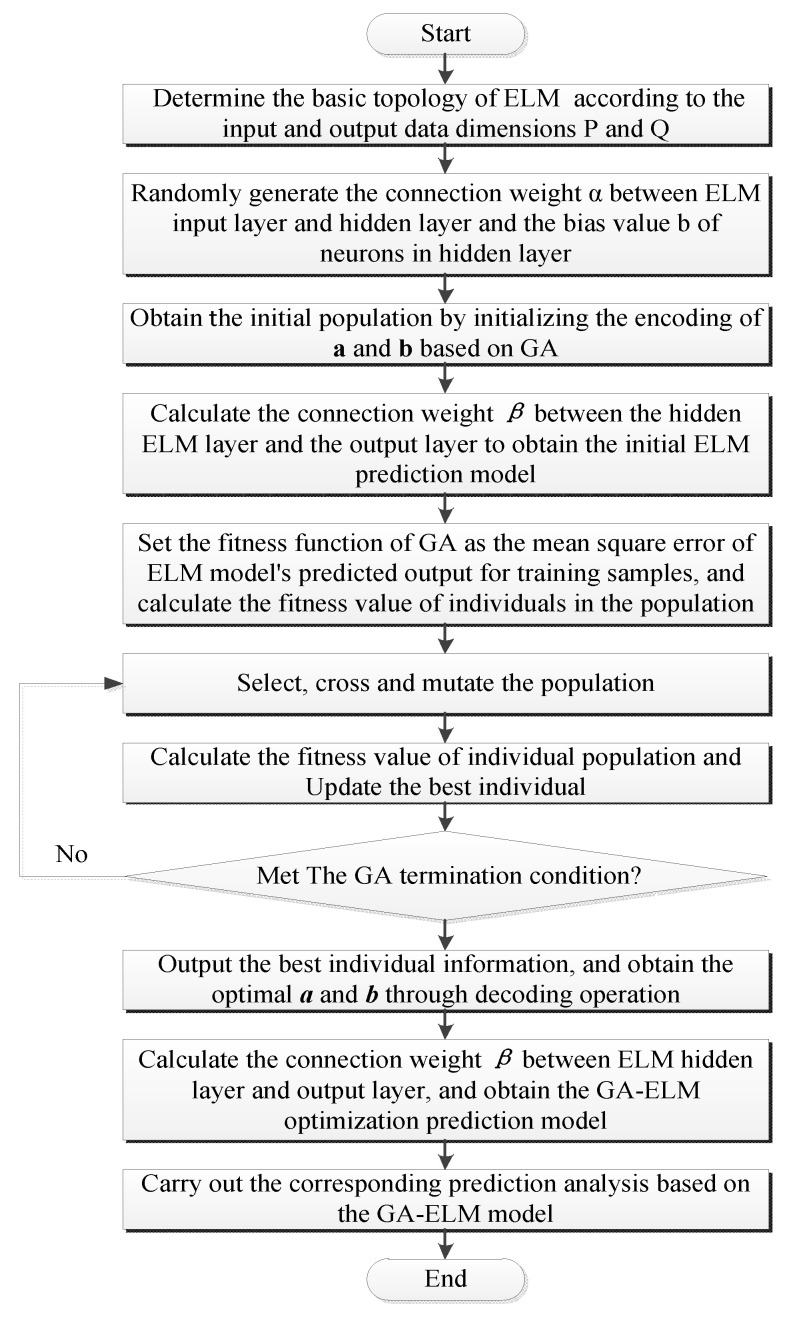
Schematic of the GA-ELM prediction model execution process.

**Figure 3 entropy-25-01343-f003:**
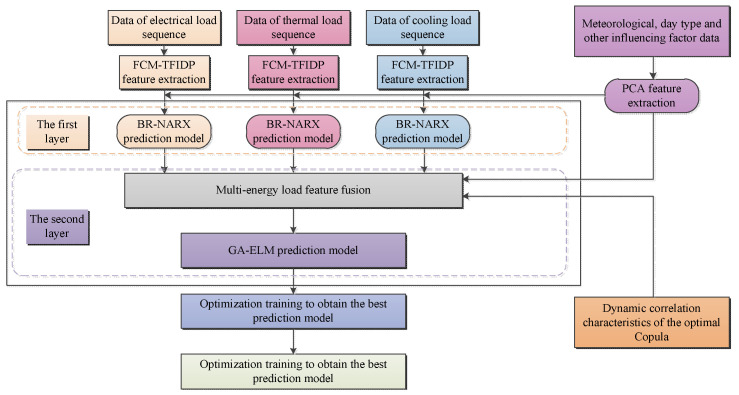
Flow-chart of the proposed multi-energy load forecasting model.

**Figure 4 entropy-25-01343-f004:**
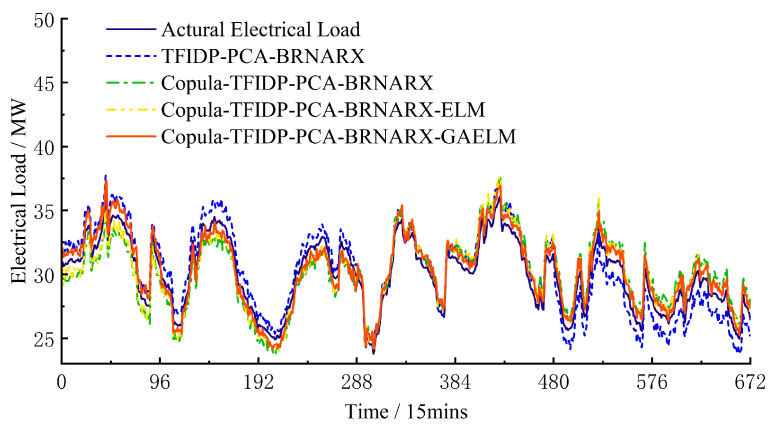
Weekly electrical load forecast results.

**Figure 5 entropy-25-01343-f005:**
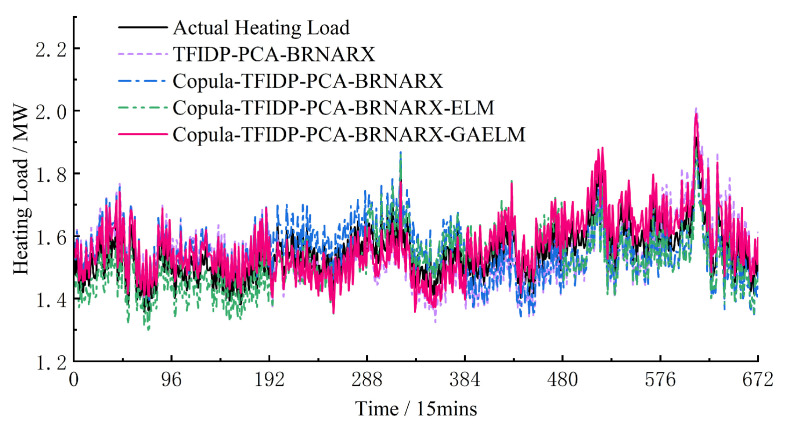
Weekly heating load forecast results.

**Figure 6 entropy-25-01343-f006:**
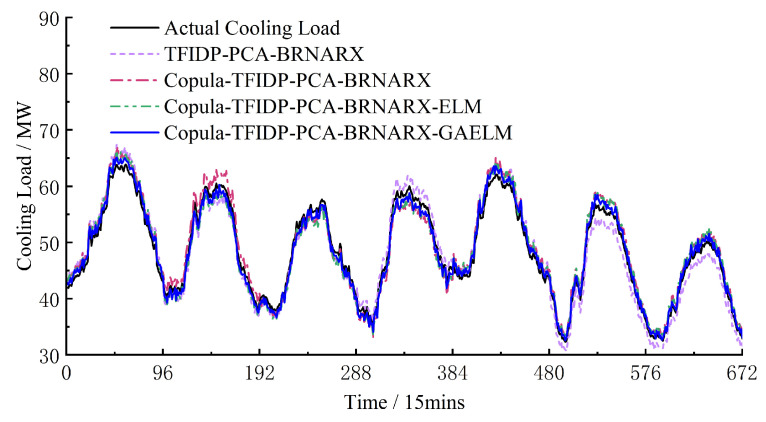
Weekly cooling load forecast results.

**Figure 7 entropy-25-01343-f007:**
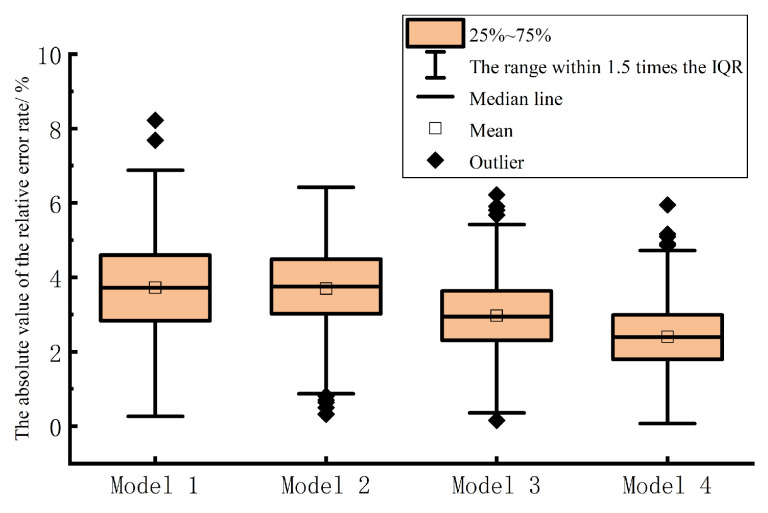
Comparison of the absolute values of relative error rates in electrical load forecasting.

**Figure 8 entropy-25-01343-f008:**
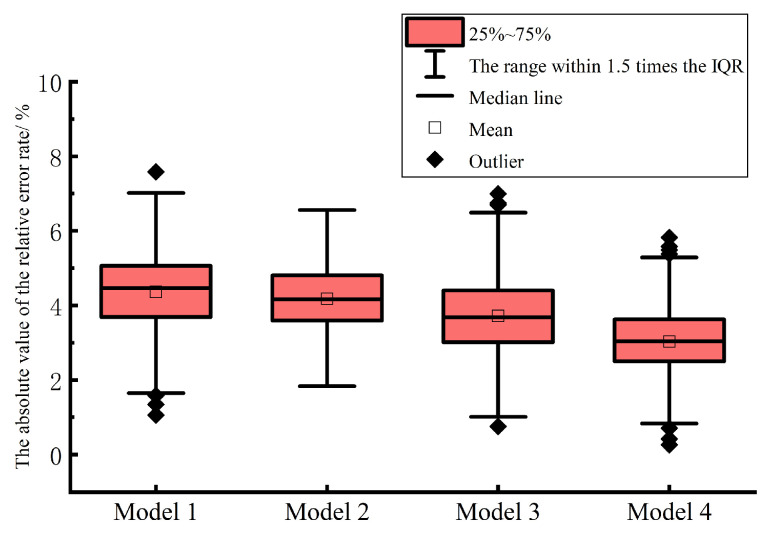
Comparison of the absolute values of relative error rates in heating load forecasting.

**Figure 9 entropy-25-01343-f009:**
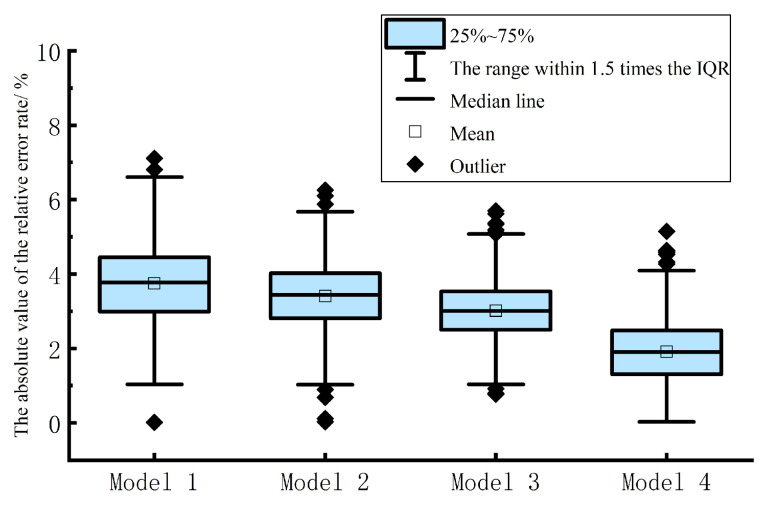
Comparison of the absolute values of relative error rates in cooling load forecasting.

**Table 1 entropy-25-01343-t001:** Performance comparison of copula functions (obtained for electrical and cooling load data) based on the AIC and maximum likelihood estimate values.

Types of Copula Function	AIC	Maximum Likelihood Estimate
Static N-copula	553.458	−311.165
Dynamic N-copula	−1725.336	851.325
Static T-copula	366.878	−197.563
Dynamic T-copula	−1935.928	937.112
Static Clayton copula	627.436	−407.601
Dynamic Clayton copula	−2817.727	1329.752
Static SJC copula	−1622.901	677.244
Dynamic SJC copula	−3284.187	1648.263

**Table 2 entropy-25-01343-t002:** Key data of the model components.

Data	The Content of the Data
Sampling interval of load data	The sampling interval is 15 min, and the daily load curve is composed of 96 load points from 00:0 to 23:45.
Factors affecting the load	(1) Meteorological information Daily temperature, humidity, air pressure, wind direction and wind speed (2) Working days, rest days and holidays information
Training set	80% of the previous data
Testing set	20% of the previous data
The reference input of the load at time t to be predicted	(1) The load at time *t* of the preceding 3 days with the same load category (considering similar days) (2) The load at time *t* − 7 to *t* − 1 (considering the relevant time) (3) Other energy load characteristic information from similar days (4) Meteorological information and day type rule information for both similar days and forecasted days

**Table 3 entropy-25-01343-t003:** Key parameters of the model components.

Model	Parameter	Parameter Settings
BR-NARX	Total number of layers	3
	Number of neurons in hidden layer	18
	Order of time delay	7
GA	Population size	40
	Number of iterations	200
	Crossover probability	0.85
	Mutation probability	0.1
ELM	Number of neurons in input layer	190
	Number of neurons in hidden layer	25
	Number of neurons in output layer	96

**Table 4 entropy-25-01343-t004:** Performance metric evaluation of the multi-energy load prediction results on weekdays.

	Evaluation Index	*E*_RMSE_ (Electrical/Heating/Cooling) (MW)	*E*_MAPE_ (Electrical/Heating/Cooling) (%)	*E*_SUMMAPE_ (%)	*Acc*_SUM_ (%)
Prediction Model					
Group 1		1.077/0.066/1.816	3.280/4.212/3.411	3.519	96.329
Group 2		1.078/0.065/1.761	3.340/4.137/3.301	3.484	96.381
Group 3		0.928/0.053/1.533	2.831/3.411/2.905	2.977	96.892
Group 4		0.747/0.047/1.101	2.268/2.962/1.998	2.299	97.544

**Table 5 entropy-25-01343-t005:** Performance metric evaluation of multi-energy load prediction results on rest days.

	Evaluation Index	*E*_RMSE_ (Electrical/Heating/Cooling) (MW)	*E*_MAPE_ (Electrical/Heating/Cooling) (%)	*E_S_*_UMMAPE_ (%)	*Acc*_SUM_ (%)
Prediction Model					
Group 1		1.390/0.078/2.044	4.811/4.751/4.596	4.713	95.199
Group 2		1.320/0.070/1.669	4.590/4.267/3.667	4.156	95.760
Group 3		0.975/0.074/1.488	3.308/4.483/3.276	3.530	96.371
Group 4		0.816/0.053/0.870	2.725/3.189/1.705	2.410	97.431

## Data Availability

Data available on request due to restrictions e.g., privacy. The data presented in this study are available on request from the corresponding author. The data are not publicly available due to the restrictions, which includes the fact that the data obtained from the City Power Authority currently do not have the authorization for public release.

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
