# Peer review of "Short-Term Prediction of Multi-Energy Loads Based on Copula Correlation Analysis and Model Fusions"

_entropy, 2023, doi:10.3390/e25091343_

Round 1
Reviewer 1 Report
The authors developed a short-term prediction method based on the copula correlation feature analysis and two-stage optimization for feature extraction. The topic is interesting. However, there are some drawbacks in the manuscript which stopped to publish in this present form.
1. The Introduction is not clear. Provide more explanation on the gap between the proposed methodology and the existing methods.
2. The main objective of the work must be written in a clearer and concise way at the end of the introduction section, discussing the necessity for the conducted work.
3. The authors should explain the studied data resolution whether used dataset is hourly basis or daily?
4. Discussion of the work is weak.
5. English should be improved. For example, the first sentence in results is problematic.
Reviewer 2 Report
- Please in the introductio better explain the theory that you are using, what is the description of this theory? Why is it used and when?
- Please in the introduction, better highlight the improvement introduced by your paper with respect to literature. Your predictive model seems to be the adding of many techniques but the reader has some difficulties to follow your description and he cannot get in which step he can find the novelty of the proposed method.
- In the introduction you say that not many works are dealing with predictive methodologies considering multi-energy vectors, so in this matter wht your method proposes with comparison to literature? Please provide the scientific gap you are willing to fill with your method.
- What type of optimizer are you considering in your method?
- In which software environment did you perform you method?
- Please improve the Table layout;
- Please improve the quality of Figure 4, 5 and 6 that are hard to be visible;
- Did you compare your analysis with other methodologies?
- From the plots of your results, the improvement of your method is not clear;
- Please give further comments on the results that come from the plots you are shown;
- You should spend more time to explain easily to the reader all the steps of your methodology since you mix many techniques in only one method.
The english of the paper is good.
Reviewer 3 Report
1. Suggest authors to add some literature review of the proposed methods, like their advantages, and the status of their usage in previous studies..
2. Strongly suggest authors to condense the part 2, Materials and methods. Authors should delete the description of other methods, meanwhile, pay more attention to the proposed Copula Correlation Analysis and how to combine models.
3. As a key part of this paper, authors should indicate the data source, such as a public dataset or a private? if public, it's better put the link there.
4. I've never seen a Table like Table 3 and 4.
5. Please enlarge figure 4-6, and make sure them clearer.
6. Besides the figures, please add the comparison between the proposed methods with other models using some other evaluation standards, such as RMSE and MAPE.
Round 2
Reviewer 1 Report
Thanks for the revision, authors improved the manuscript well.
Author Response
The authors appreciate the reviewer.
Reviewer 2 Report
The authors improved the introduction highlighting the novelty proposed by the paper.
Although the quality of the Fig. 4-5 and 6 is still very low they are difficult to read even by zooming.
Also the size of Fig.3 can be increased.
Please improve the quality of these Figures.
Reviewer 4 Report
All the suggestions have been addressed.
Author Response
The authors appreciate the reviewer.